# Non-fixation versus fixation of mesh in laparoscopic transabdominal preperitoneal repair of inguinal hernia: A systematic review and meta-analysis of randomized controlled trials

**ChenXin Zhang**[1], **Jia Li**[2], **HaiJin Suo**[1], **JianPing Bai**[1]*

1 Department of General Surgery, The 983rd Hospital of Joint Logistic Support Force of PLA, Tianjin, China,
2 Department of Gastroenterology, The 983rd Hospital of Joint Logistic Support Force of PLA, Tianjin, China

* baijp1971@163.com

**Data Availability Statement:** All relevant data are within the manuscript and its Supporting Information files.

## Abstract

### Purpose

The impact of non-fixation of mesh in transabdominal preperitoneal (TAPP) inguinal hernia repair has not been fully assessed. The aim of this meta-analysis was to comprehensively compare the clinical outcomes of non-fixation and fixation of mesh in TAPP to determine whether non-fixation could affect the outcomes.

### Methods

PubMed, Embase and CENTRAL were searched for studies on TAPP repair of inguinal hernia and mesh fixation published up to June 2023. The literature search was completed on June 22, 2023. Randomized controlled trials that compared perioperative outcomes between mesh fixation and non-fixation without using self-gripping mesh were included. The primary outcome measures were recurrence and evaluation of postoperative pain, while secondary outcome measures included time to normal activity, infection rate and formation of seroma. Subgroup analyses and sensitivity analysis were also conducted.

### Results

Six randomized controlled trials were included, involving 679 patients who underwent TAPP with non-fixation and 964 patients with fixation of mesh. There was no significant difference in recurrence between the two groups (RR: 0.83; 95% CI, 0.29–2.39, P = 0.73). The non-fixation group had less pain than the fixation group at 6 months postoperatively (MD: -0.16; 95% CI, -0.23–-0.10, P < 0.0001). Additionally, there was no significant difference in the time to return to normal activity or rates of infection or seroma formation between the two groups (MD: -4.95; 95% CI, -11.36–1.45, P = 0.13; RR: 1.18; 95% CI, 0.39–3.62, P = 0.77; RR: 0.94; 95% CI, 0.63–1.40, P = 0.75).

**Funding:** The author(s) received no specific funding for this work.

**Competing interests:** The authors have declared that no competing interests exist.

## Conclusion

Based on the current evidence, non-fixation without using self-gripping mesh may not affect the efficacy of TAPP. It does not increase recurrence rate and may result in less postoperative pain in inguinal hernia with small hernia defect (less than 3cm).

## Introduction

Inguinal hernia is a common disease in general surgery, with over 20 million patients worldwide undergoing repair surgery annually [1]. For symptomatic inguinal hernias, mesh repair is recommended for treatment. Surgical procedures for inguinal hernia repair are primarily classified as open repair and laparoscopic mesh repair. Despite open mesh repair already having low recurrence rate [2], laparoscopic mesh repair has gained popularity in recent years due to its advantages such as well recognition of groin anatomy, convenience in dealing with bilateral hernias and good cosmetic effect.

Laparoscopic totally extraperitoneal (TEP) and transabdominal preperitoneal (TAPP) repair are two most commonly used laparoscopic mesh repair techniques, both of which are based on the theory of myopectineal orifice. The primary distinction between TEP and TAPP is the approach to enter preperitoneal space. In TAPP, preperitoneal space is entered by dissecting parietal peritoneum through peritoneal cavity, while in TEP, the entering of peritoneal cavity and subsequent peritoneal closure are avoided. It is widely accepted that standard dissection of preperitoneal space [3], proper placement and fixation of the mesh are key points in both TEP and TAPP technique. However, there is still some controversy regarding mesh fixation in laparoscopic mesh repair. Advocates argue that mesh fixation can prevent mesh migration, which is believed to be an essential cause of hernia recurrence after laparoscopic repair [4]. While opponents contend that mesh fixation may increase the risk of nerve injury, potentially leading to postoperative pain including chronic groin pain (CGP) [5]. Several meta-analyses of randomized controlled trials (RCTs) have already compared the perioperative outcomes between non-fixation and fixation of the mesh in laparoscopic inguinal hernia repair [6–10]. They all demonstrated that non-fixation did not increase the risk of hernia recurrence. However, there were inconsistent results regarding postoperative pain. It is important to note that these systematic reviews were mainly focusing on TEP technique and there was only one review exploring the effects of non-fixation of mesh in TAPP separately [10]. Moreover, in that meta-analysis of Riemenschneider et al. [10], studies using self-gripping mesh which was considered as an atraumatic mesh fixation method were included. And only hernia recurrence and CGP between non-fixation and fixation of the mesh were investigated. Therefore, the influence of mesh non-fixation in TAPP remains not sufficiently concluded. The object of our present review was to comprehensively compare the clinic outcomes of non-fixation and fixation of the mesh in TAPP inguinal hernia repair to determine whether non-fixation could affect the outcomes.

## Materials and methods

### Search strategy

PubMed, Embase and Cochrane Central Register of Controlled Trials (CENTRAL) were searched for studies about TAPP repair of inguinal hernia and mesh fixation published up to June 2023. All searches were finished at June 22, 2023. The following medical subject headings

(MeSH) terms: "Hernia, Inguinal", "Hernia, Femoral" and relevant free words, such as "groin hernia", "transabdominal preperitoneal", "TAPP", "fixation", were used for search. The complete search strategies are shown in S1 Appendix. The references of relevant reviews were also reviewed to identify additional studies.

## Inclusion of studies

Studies of randomized controlled trials which compared perioperative outcomes between mesh fixation and non-fixation of TAPP repair of inguinal hernia were included. TAPP repair was done according to the standard way [11]. Non-fixation of mesh was defined as leaving the mesh in preperitoneal space without fixing the mesh to around structures. Using self-gripping mesh was not considered as non-fixation. Fixation of mesh was defined as using traumatic methods (suturing, stapler, tacker) or atraumatic method (adhesive glue) to fix the mesh in the preperitoneal space. Non-English articles were excluded.

## Data extraction

Data extraction was conducted by two independent authors (CXZ and JL) with a pre-designed spreadsheet. The spreadsheet was created according to the Cochrane's recommendations for intervention reviews. The following information was needed in the spreadsheet: 1. study-related data (first author, publication year, country of origin, publication journal, study period, follow-up time, method of randomization), 2. baseline of study population (study size, age, gender, weight, laterality of hernia, hernia defect size, history of recurrence), 3. procedure details (surgeon experience, type and size of mesh, method of fixation, use of antibiotic, drainage), 4. Primary outcomes (recurrence, evaluation of postoperative pain, chronic groin pain) and secondary outcomes (operation time, hospital stay, time to normal activity/work, infection, formation of seroma, Intraoperative complication, cost). Disagreements during data extraction were discussed until a consensus was made. The final data was checked by author CXZ.

## Assessment of included study quality

The quality of included studies was assessed by two authors (CXZ and HJS) using the revised Cochrane risk of bias tool for randomized trials [12]. The Cochrane's tool assesses the risk of bias of RCTs in five domains: bias arising from the randomization process, bias due to deviations from intended interventions, bias due to missing outcome data, bias in measurement of the outcome, and bias in selection of the reported result. Each item is classified as low risk, some concerns or high risk. Discrepancies in risk of bias assessment were resolved by consulting to author (JPB).

## Statistical analysis

For dichotomous outcome variables (recurrence, chronic groin pain, seroma, infection and intraoperative complication), the risk ratio (RR) was calculated as summary measure. The RR is a ratio of the risk of an adverse event in the non-fixation group compared to that in fixation group. For continuous outcomes (operation time, postoperative pain, time to normal activity/ work), the mean difference (MD) between the non-fixation group and fixation group was calculated as summary measure. If mean and (or) standard difference (SD) of interested outcome were not provided, they were tried to calculated through methods described by Hozo et al [13]. All data synthesis was performed using Review Manager 5.3 software (Cochrane Collaboration, Oxford, England). Fixed effect model or random effect model was used during data

synthesis according to heterogeneity among the included studies. Heterogeneity was assessed by Q test and Higgins I2 statistic. Random effect model was used if p<0.10 or I2>50%.

Subgroup analyses were conducted according to mesh fixation method and hernia laterality to assess the robustness of results. Sensitivity analysis was also performed by repeating the analysis following excluding one study at a time. Publication bias could not be evaluated as there were totally fewer than 10 studies included.

## Results

### Study selection

Search and selection procedure are shown in Fig 1. Each step of selection was conducted by two authors (CXZ and HJS) independently. Disagreements were resolved by discussion. Through systematically searching in PubMed, Embase and Cochrane database, 592 records were retrieved. Adding one additional article from bibliography, there were totally 593 records for further review. After removing duplicates, titles and abstracts of 463 records were screened. Then there were 19 articles remained after title and abstract screening. Full texts of the 19 articles were intent to retrieve for further review. Four conference abstracts and 2 articles of which the full texts were not possible to get were excluded. Two articles in Russian and 5 non-randomized trial articles were also excluded. Finally, six prospective, randomized controlled trials were included for meta-analysis. Details can be seen in S1 File.

### Description of included studies

The characteristics of the six included studies are presented in Table 1. There were a total of 1643 patients, of which 679 patients underwent TAPP surgery with non-fixation of mesh and

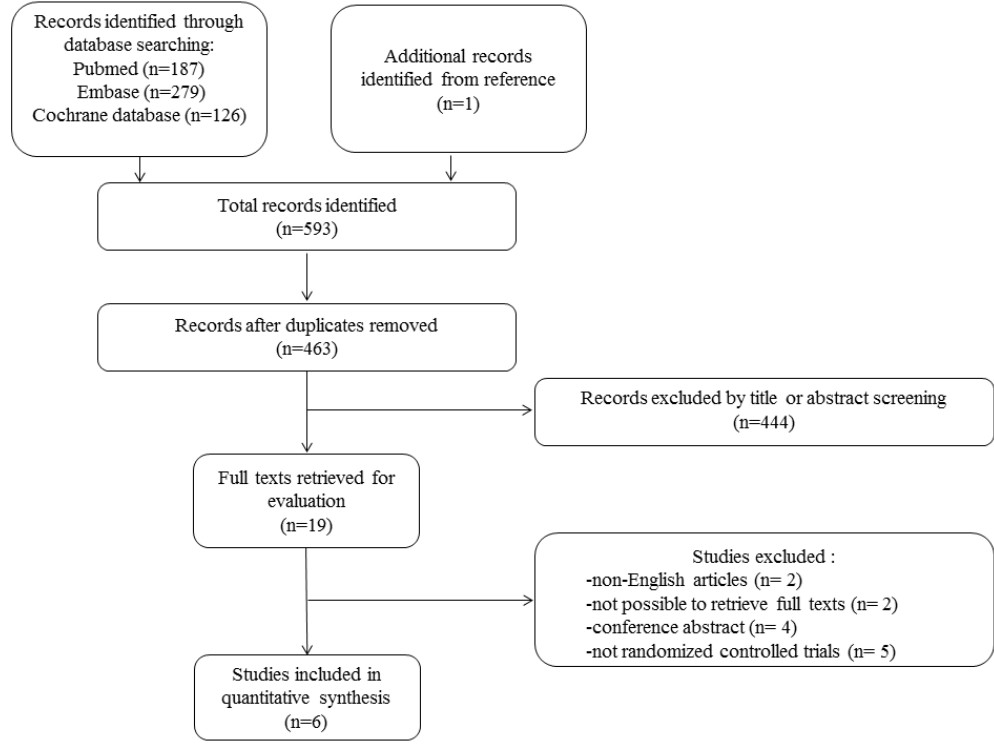

**Fig 1. The flowchart of study selection.**

**Table 1. Characteristics of included studies.**

| Author | Year | Country | No. of patients | Study period | Method of randomization | Blind or not | Hernia details | Hernia defect | History of recurrence | Mean follow-up time (months) |
|---|---|---|---|---|---|---|---|---|---|---|
| Smith [15] | 1999 | Australia | 502 | Started in January 1995 | NM | NM | unilateral or Bilateral inguinal hernia | NM | NM | 17 |
| Li [17] | 2017 | China | 100 | July of 2015 to July of 2016 | Randomization sequence list | NM | Primary unilateral oblique inguinal hernia | Mean hernia defect diameter | no | NFG 11.5 |
| | | | | | | | | NFG 2.6 ± 0.7 (1.0–3.9)cm | | FG 11.2 |
| | | | | | | | | FG 2.5 ± 0.8 (1.0–3.8)cm | | |
| Kalidarei [14] | 2019 | Iran | 80 | April 2017 to March 2018 | Computer randomization | NM | unilateral or Bilateral inguinal hernia | NM | no | 6 |
| Habeeb [19] | 2020 | Egypt | 798 | June 2013 to June 2018 | Random sequence generator | NM | Primary unilateral Indirect inguinal hernia | NM | no | 18 |
| Azevedo [16] | 2022 | Brazil | 63 | November 2016 to November 2019 | Computer randomization | double-blind | Primary unilateral inguinal hernia | Defect size (≤1.5, ≤3, >3)cm | no | 24 |
| | | | | | | | | NFG (1/18/2) | | |
| | | | | | | | | FG (2/36/4) | | |
| Meshkati [18] | 2023 | Iran | 100 | March 2021 to August 2021 | Permuted block randomization | single-blind | Primary unilateral inguinal hernia | NM | no | 6 |

NM not mentioned, FG fixation group, NFG non-fixation group

964 patients with fixation of mesh. Except for the additional article from reference, the other five studies were all conducted in the last decade. In the study of Kalidarei et al. [14] and the study of Smith et al. [15], there were 16 patients and 34 patients of bilateral inguinal hernia, respectively. Two studies [16, 17] described the size of hernia defect and there was no significant difference of the size of hernia defect between non-fixation group and fixation group. The mean follow-up time ranged from 6 to 24 months.

The details of procedure are shown in Table 2. Except for two studies [17, 18] in which all the surgeries were done by one experienced surgeon, the other four studies had a surgical team to perform the TAPP surgery. A kind of lightweight 3D mesh was used in the study of Li et al. [17] and normal polypropylene mesh was used in the remaining five studies. In all included studies the size of mesh was at least 15cm × 10cm. Suture, tacker, stapler and glue were mesh fixation methods used in the six included studies. Three studies [14, 16, 19] had used two methods for mesh fixation. Two studies [18, 19] mentioned the use of prophylactic antibiotic. None of the studies mentioned the use of drainage. One study [19] reported that 15 cases were converted to open surgery, but there were no more details.

## Assessment of study quality

The risk of bias assessment of the included studies for recurrence is presented in Table 3. Overall, two [14, 19] out of the six trials were considered as 'some concerns' because they did not report their process of concealment. The overall risk of one trial [15] was high as the process of randomization was not reported and the rate of lost follow-up was high, in addition with some outcomes were assessed by patient themselves. The results of bias assessment of other outcomes are shown in S2 File.

**Table 2. Procedure details.**

| Author | Surgeon experience | Type and size of mesh | Method of fixation | Use of prophylactic antibiotic | Convert to open surgery | Postoperative complication |
|---|---|---|---|---|---|---|
| Smith [15] | Surgeons performed more than 200 TAPP repairs | polypropylene mesh 15cm × 10cm | EMS Ethicon stapler | NM | NM | Seroma, mesh infection, wound infection, urinary retention |
| Li [17] | One surgeon with an associate director title | Easyprosthes lightweight 3D mesh 12 cm × 16 cm | Covidien stapler | NM | no | Seroma, wound infection |
| Kalidarei [14] | One surgical team | Prolene mesh 15 cm × 10 cm | suture or spiral tacks | NM | no | Seroma, wound infection, urinary retention, neuralgia |
| Habeeb [19] | Surgeons performed more than 100 repairs | polypropylene mesh 15cm × 10 cm | Tacker or Histoacryl | third generation cephalosporins used for 24h after operation | 15 cases converted | Seroma, wound infection, testicular atrophy, mesh infection |
| Azevedo [16] | One surgical team | polypropylene mesh at least 15cm × 12 cm | n-butyl-2-cyanoacrylate or stapler | NM | no | seroma |
| Meshkati [18] | an experienced surgeon in the field of laparoscopic repair | Polypropylene mesh 15cm × 13 cm | absorbable tacks | cefazoline single dose (2g) pre-operatively | no | Urinary retention |

NM not mentioned

## Outcomes

**Primary outcomes.** *Recurrence.* Recurrence rate was reported as an outcome in all included studies. The total recurrence rate in non-fixation group and fixation group were 0.6% and 0.7%, respectively. The pooled analysis did not demonstrate any significant difference in recurrence rate between the two groups (RR: 0.83; 95% CI, 0.29–2.39, P = 0.73). Heterogeneity among the included studies was low (I2: 8%, P = 0.35) (Fig 2).

*Postoperative pain.* Visual Analog Scale (VAS) pain scale, in which pain score is defined from 0 (no pain) to 10 (maximum pain), was used for postoperative pain evaluation in five included studies [14, 16–19]. Among them, the study of Azevedo et al. [16] also used McGill pain questionnaire for pain evaluation. Four [14, 16–18] of the five studies reported VAS pain score data of different time. Azevedo et al. [16] demonstrated that at six different time periods (first postoperative day, between Day 7 and Day 15, after 3 and before 6 months, after 1 year, after 1 year and 6 months and after 2 years) there were all no significant difference in pain score of VAS scale between non-fixation group and mesh fixation group. In the study of Li et al. [17], VAS pain scores of 2 days, 3 months, and 6 months postoperatively of non-fixation group were all significantly lower than those in fixation group. Data synthesis could only be

**Table 3. Risk of bias of included studies for recurrence.**

| Domain of bias | Azevedo [16] | Habeeb [19] | Kalidarei [14] | Li [17] | Meshkati [18] | Smith [15] |
|---|---|---|---|---|---|---|
| bias arising from the randomization process | 0 | 1 | 1 | 0 | 0 | 1 |
| bias due to deviations from intended interventions | 0 | 0 | 0 | 0 | 0 | 0 |
| bias due to missing outcome data | 0 | 0 | 0 | 0 | 0 | 1 |
| bias in measurement of the outcome | 0 | 0 | 0 | 0 | 0 | 1 |
| bias in selection of the reported result | 0 | 0 | 0 | 0 | 0 | 0 |
| Overall risk of bias | 0 | 1 | 1 | 0 | 0 | 2 |

0 = low risk, 1 = some concerns, 2 = high risk

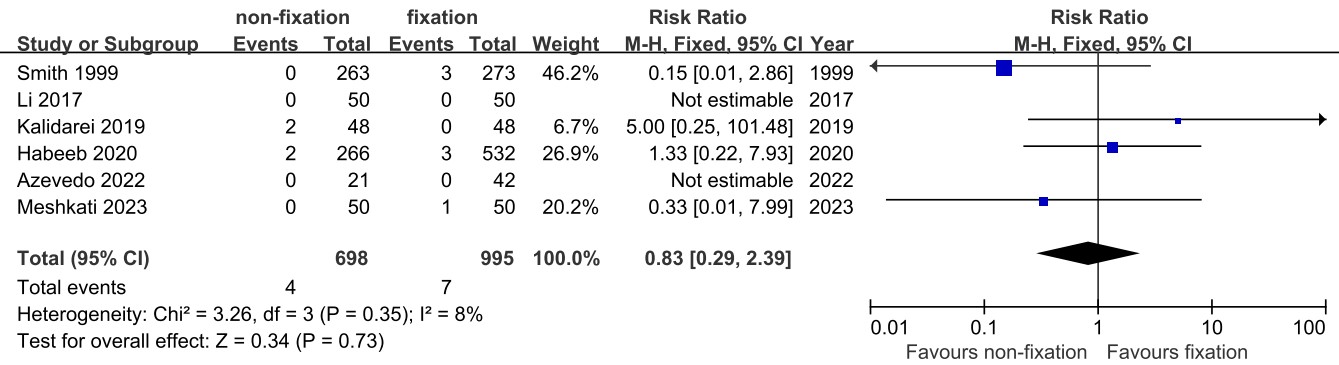

**Fig 2. Meta-analysis of recurrence between non-fixation and fixation.**

done for postoperative pain at 6 months as three studies [16–18] reported VAS pain score of 6 months postoperatively. The analysis showed pain score at 6 months postoperatively of non-fixation group was significantly lower than that of fixation group (MD: -0.16; 95% CI, -0.23–-0.10, P < 0.0001). Between-study heterogeneity was moderate (I2: 44%, P = 0.17) (Fig 3).

*Chronic groin pain*. Chronic pain was reported as an outcome in four studies [14, 16, 18, 19]. Three [14, 16, 18] of them showed there was no chronic pain in both non-fixation group and fixation group. The other study [19] showed the chronic pain rate were 1.9% and 13.5% in non-fixation group and fixation group respectively. However, only the study by Azevedo et al. [16] and the study by Habeeb et al. [19] gave a clear definition of chronic pain (chronic pain was defined as persistent pain lasting over 3 months). The available data did not allow us to conduct quantitative synthesis on chronic pain.

**Secondary outcomes.** *Wound and mesh infection*. Four studies [14, 15, 17, 19] provided data of wound and mesh infection. There was no significant difference in infection rate between non-fixation group and fixation group (RR: 1.18; 95% CI, 0.39–3.62, P = 0.77). The between-study heterogeneity was low (I2: 24%, P = 0.25) (Fig 4).

*Seroma*. Data of seroma formation were available from five studies [14–17, 19]. The total rate of seroma formation in non-fixation group and fixation group were 7.2% and 5.9%, respectively. The pooled analysis showed there was no significant difference of seroma formation rate between the two groups (RR: 0.94; 95% CI, 0.63–1.40, P = 0.75). There were moderate heterogeneity among the included studies (I2: 38%, P = 0.17) (Fig 5).

*Time to normal activity*. Time to normal activity was reported by three studies [14, 15, 18]. There was no significant difference in the time taken to normal activity between non-fixation group and fixation group (MD: -4.95; 95% CI, -11.36–1.45, P = 0.13). There was a high level of heterogeneity among the studies (I2: 98%, P<0.00001) (Fig 6).

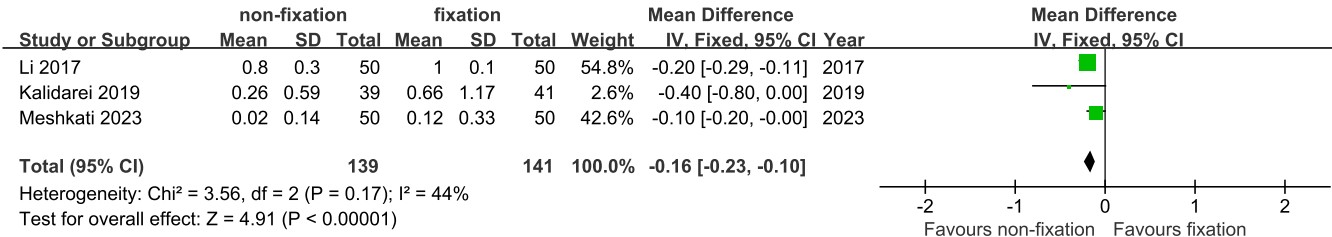

**Fig 3. Meta-analysis of postoperative pain at 6 months between non-fixation and fixation.**

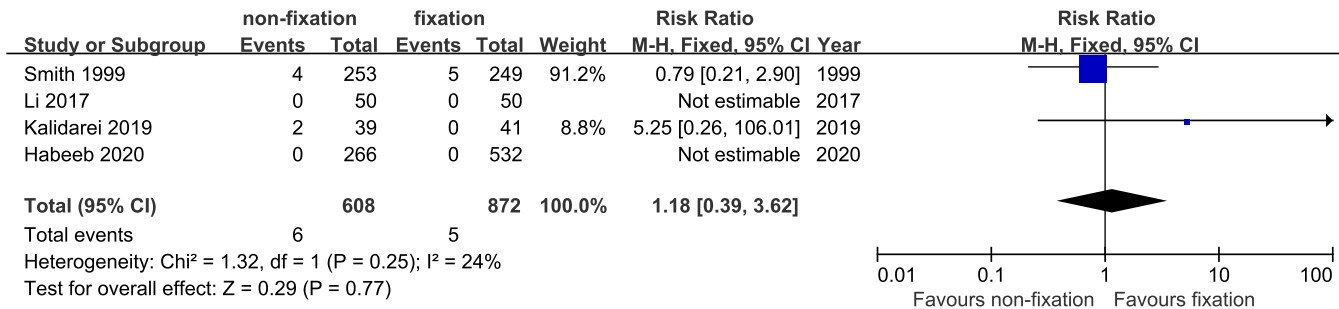

**Fig 4. Meta-analysis of infection between non-fixation and fixation.**

*Operation time.* Three studies [14, 15, 17] reported data of operation time. In the study of Li et al. [17], they demonstrated the mean operation time of non-fixation (50.1 ± 10.3 min) group was significantly reduced compared to that of the fixation group (60.5 ± 12.2 min). While Kalidarei et al. [14] reported there was no significant difference of operation time between the two groups(non-fixation 70.2±0.44 min vs fixation 73.8±0.54 min, P = 0.585). The other study of Smith et al. [15] only reported mean value of operation time but without other information. The pooled analysis could not be conducted.

*Hospital stay.* Data of hospital stay was reported by two studies [14, 19]. Both of them reported the time of hospital stay of non-fixation group was significantly reduced compared to that of the fixation group.

*Cost of treatment.* Data of cost of treatment was available in only one study. In that study by Li et al. [17], they reported hospitalization expense was significantly lower in non-fixation group (10,560 ± 160 yuan) than that in fixation group (14,280 ± 320 yuan).

**Sub-group analysis.** In the subgroup analysis of fixation method, there were no significant difference in recurrence rate, infection rate and seroma formation between non-fixation group and group of traumatic mesh fixation(tacker, stapler, or suture) (RR: 0.87; 95% CI, 0.28–2.68, I2: 16%; RR: 1.18; 95% CI, 0.39–3.62, I2: 24%; RR: 1.18; 95% CI, 0.78–1.79, I2: 0%, respectively). In the subgroup analysis for primary unilateral hernia, the results were consistent with the primary analysis. Mesh non-fixation did not differ significantly in recurrence rate and seroma formation compared to mesh fixation (RR: 0.90; 95% CI, 0.20–4.00, I2: 0%; RR: 0.56; 95% CI, 0.14–2.20, I2: 52%).

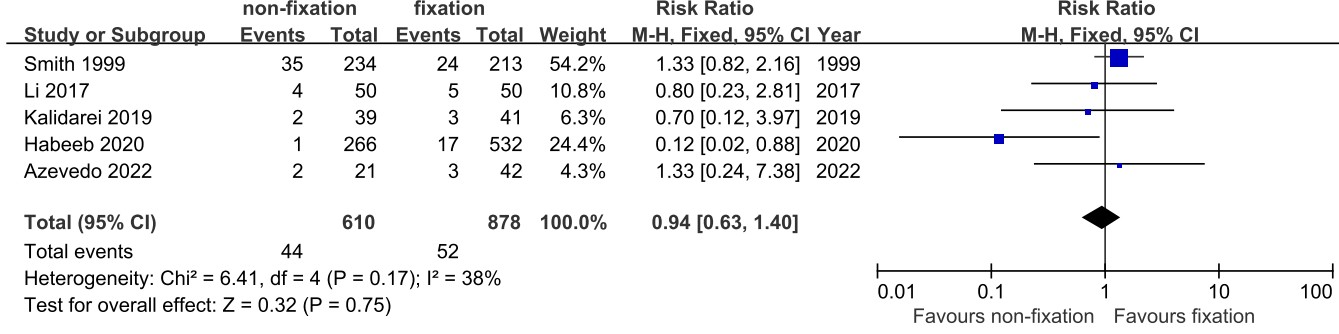

**Fig 5. Meta-analysis of seroma formation between non-fixation and fixation.**

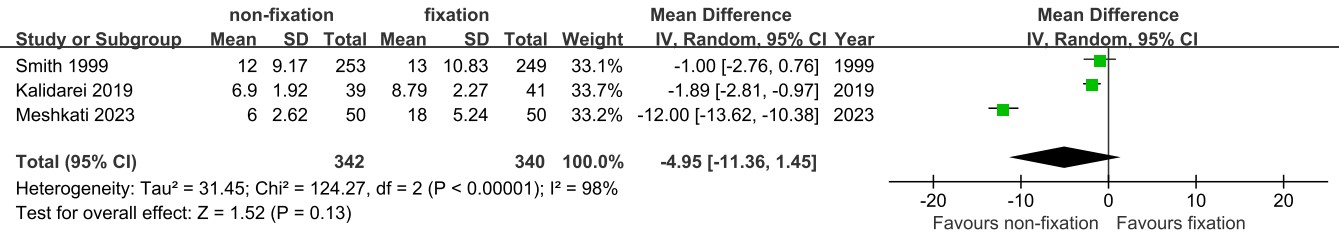

**Fig 6. Meta-analysis of time to normal activity between non-fixation and fixation.**

**Sensitivity analysis.** Sensitivity analyses were carried out by excluding one study each time. Recurrence, infection and seroma formation were evaluated. The direction of pooled effect size all remained unchanged. Sensitivity analyses for other outcomes could not be conducted due to unavailable data.

## Discussion

In this meta-analysis, the clinic outcomes of non-fixation and fixation of the mesh in TAPP were compared. There were six RCT trials included, involving 679 patients who underwent non-fixation and 964 patients who underwent fixation of the mesh. To our knowledge, this is the first meta-analysis that compares non-fixation and fixation in TAPP inguinal hernia repair without using self-gripping mesh. We exclude studies that used self-gripping mesh as typically using self-gripping mesh is considered as an atraumatic fixation method. Self-gripping mesh is designed to increase adhesion, which makes it different from normal polypropylene mesh. There may be potential bias when studies that use self-gripping mesh and normal mesh are combined for analysis. Additionally, considering the cost of treatment, there is a higher prevalence of utilizing normal polypropylene mesh. Excluding studies using self-gripping mesh could make our results more generalizable.

Recurrence after laparoscopic inguinal hernia repair has been widely discussed. Various factors related to patients and surgical techniques have been found to be associated with hernia recurrence [20]. Among these factors, fixation of mesh has garnered considerable attention. One of our primary outcomes showed non-fixation of mesh had comparable recurrence rate comparing to fixation of mesh in TAPP inguinal hernia repair. Our subgroup analyses and sensitivity analyses regarding to recurrence further confirmed the result. This finding is in line with the result of a previous meta-analysis [9] which did the same comparison in TEP. Besides, a similar conclusion was demonstrated by a retrospective study with large sample size. In that study, a total of 11228 male patients diagnosed with primary unilateral inguinal hernia underwent TAPP repair and were followed up for 1 year [21]. There were no significant difference of recurrence between non-fixation and fixation in both unadjusted analysis and multivariable analysis. Traditionally, mesh fixation was believed to prevent mesh displacement which was thought to contribute to hernia recurrence. However, studies investigated the movement of mesh after laparoscopic inguinal hernia repair had demonstrated that non-fixation may not cause mesh displacement. Claus et al. [22] measured the displacement of mesh after laparoscopic inguinal hernia repair by radiological examination of the clips that placed with mesh during surgery. They found no significant difference of displacement of mesh between non-fixation group and fixation group. Another study of Choy et al. [23] investigated the movement of unfixed mesh following hip flexion and extension immediately postoperatively on the operating table by relaparoscopy. They found the movement of well-placed unfixed mesh was minimal. These studies suggest that fixation of the mesh in laparoscopic inguinal hernia repair

may be omitted when preperitoneal space is well dissected and the mesh is properly placed. Nevertheless, it is important to mention that the follow up duration of the included trials in the present review was relatively short. Out of the six trials, only three [15, 16, 19] had a follow up period more than 12 months. Recurrence after inguinal hernia repair operation is usually considered as a long-term complication, with an increased likelihood in older patients. The limited duration of follow-up may not be sufficient for detecting recurrence.

Postoperative pain is another focused issue in inguinal hernia repair surgery. The measurement scales and time period for postoperative pain evaluation usually differ among studies, which causes difficulties in comparing and integrating data. In the present review, only data of 6 months after repair surgery could be integrated. The result showed non-fixation of mesh had less pain than fixation of mesh at 6 months postoperatively. In the meta-analysis of Sahebally et al. [9], they reported that non-fixation was associated with significantly less pain at 24h postoperatively compared to fixation in TEP. In another meta-analysis, Sajid et al. [7] demonstrated that the post-operative pain was comparable between non-fixation and fixation. However, they did not mention the exact postoperative time period for evaluation. Although those results were of different time points, they all demonstrated non-fixation of mesh had less or comparable postoperative pain comparing to fixation of mesh after laparoscopic inguinal hernia repair surgery. CGP is most commonly defined as postoperative pain in the groin region lasting for at least three months. It is an important outcome which affects the quality of life after inguinal hernia repair surgery [24]. There were only two trials in the present review giving a clear definition of CGP. Due to insufficient data, pooled analysis of CGP could not be conducted. The previous meta-analysis conducted by Sajid et al. [7] demonstrated that risk of developing CGP was similar between non-fixation and fixation in laparoscopic inguinal hernia repair. However, at least one study [15] included in their combined calculation did not give an accurate definition of CGP. Recently, CGP was recommended to be defined as moderate pain affecting daily activities more than three months postoperatively [1]. Using this new definition will make the evaluation of CGP more clinically relevant and may guide practice. The cause of postoperative pain is complicated. It is thought that nerve injury and continued inflammation at surgical site are two main reasons of acute and chronic postoperative pain. Traumatic mesh fixation methods, such as tacks, staples, and suture, may possess both of the two reasons. And this may explain the results of both present and former meta-analyses, as traumatic fixation methods were predominantly used in the trials included in the pooled analysis. Among trials included in our present review, Azevedo et al. [16] and Habeeb et al. [19] used fibrin glue for fixation in some patients of the fixation group. Fibrin glue, as an atraumatic mesh fixation method, has been reported to cause less postoperative pain compared to staple fixation in TAPP repair [25, 26]. Subgroup analysis comparing non-fixation versus fibrin glue could not be conducted in present review due to a lack of available data. In a retrospective study of internet-based Herniamed Registry, Niebuhr et al. [27] found there was no difference in pain at rest and on exertion between non-fixation and glue fixation in TAPP inguinal hernia repair after one year follow up. They also demonstrated no difference in chronic pain requiring treatment between non-fixation and mesh fixation with fibrin glue. It seems that glue fixation has comparable postoperative pain when compared to non-fixation. Further randomized controlled trials are required to confirm these findings.

It is still uncertain whether non-fixation of mesh in laparoscopic inguinal hernia repair could improve the return to daily activities. A previous meta-analysis by Eltair et al. [8] demonstrated that the time for return to daily activities was comparable between non-fixation and fixation of mesh in laparoscopic inguinal hernia repair. Their pooled analysis included data from four RCTs, three of which were of TEP and only one trial was of TAPP. The current meta-analysis demonstrated a similar result in TAPP. However, it is important to note that

there were only three trials included in the pooled analysis and the between-study heterogeneity was high. Therefore, the result should be interpreted with caution. It is always challenging to study the convalescence of hernia repair surgery. As not only factors related to surgery but also those related to social-economy, such as patient motivation, culture, and even the insurance, could influence the recovery process after hernia repair [28]. Furthermore, there is still no consensus on recommendation for convalescence duration following inguinal hernia repair [29]. All these factors may complicate the study of convalescence.

The present meta-analysis did not show any difference between non-fixation and fixation in the occurrence of surgery-related infections or the formation of seroma. The results were in agreement with those of previous meta-analyses. Laparoscopic inguinal hernia repair is a challenging technique with a steep learning curve. The European Hernia Society recommends surgeons to perform at least 50–100 procedures to overcome the learning curve and reduce the rate of complications [30]. It appears that the surgeon's experience is the main determinant of complications in laparoscopic hernia repair surgery.

There are several limitations of the present review. First, we were unable to evaluate chronic groin pain, operation time, hospital stay, or costs due to lack of available data from the included trials. Of all these, chronic groin pain is an important outcome measure for assessing the safeness of laparoscopic hernia repair surgery. Moreover, as far non-fixation of mesh is recommended for inguinal hernia with the diameter of defect less than 4cm by The International Endohernia Society guidelines [11]. However, only two included trials reported the size of hernia defect and most cases had a defect size less than 3cm. Therefore, the efficacy and safety of non-fixation in inguinal hernia with large defect still needs to be clarified. Finally, in this review, the mean and standard deviation of some continuous outcome variables, which were presented as median and interquartile range, were calculated by a widely accepted method. It should be noted that errors may occur during the calculation process, potentially introducing bias into the results.

## Conclusion

Non-fixation may not affect the efficacy of TAPP based on the current evidence. It does not increase recurrence rate and may result in less postoperative pain compared to mesh fixation in inguinal hernia with small hernia defect (less than 3cm). Well-designed RCTs using TAPP technique with large sample size and adequate follow up are still required to confirm these findings and validate the influence of non-fixation on chronic groin pain.

## Supporting information

**S1 Checklist. PRISMA checklist.**
(DOCX)

**S1 Appendix. Search strategy.**
(DOCX)

**S1 File. Search result.**
(DOCX)

**S2 File. Bias assessment.**
(DOCX)

**S1 Data. Extracted data for meta-analysis.**
(DOCX)

## Author Contributions

**Conceptualization:** ChenXin Zhang.

**Data curation:** HaiJin Suo.

**Investigation:** HaiJin Suo.

**Methodology:** ChenXin Zhang, Jia Li.

**Project administration:** ChenXin Zhang.

**Software:** ChenXin Zhang.

**Supervision:** JianPing Bai.

**Validation:** HaiJin Suo, JianPing Bai.

**Writing – original draft:** ChenXin Zhang.

**Writing – review & editing:** JianPing Bai.

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
