## [Decision Letter · Decision Letter 0]

22 Jul 2024

PONE-D-24-23572Non-fixation versus fixation of mesh in laparoscopic transabdominal preperitoneal repair of inguinal hernia: a systematic review and meta-analysis of randomized controlled trialsPLOS ONE

Dear Dr. bai,

Thank you for submitting your manuscript to PLOS ONE. After careful consideration, we feel that it has merit but does not fully meet PLOS ONE’s publication criteria as it currently stands. Therefore, we invite you to submit a revised version of the manuscript that addresses the points raised during the review process.

We look forward to receiving your revised manuscript.

Kind regards,

Lovenish Bains, MS, FNB, FACS, FRCS (Glas), FICS, FIAGES

Academic Editor

PLOS ONE

Journal Requirements:

Additional Editor Comments:

The manuscript presents a meta-analysis comparing non-fixation versus fixation of mesh in laparoscopic transabdominal preperitoneal hernia repair. It is generally clear to read, though some subheadings and references need formatting adjustments as per journal standards. Statistical methods need clarification regarding cases converted to open surgery, and consistency in pain evaluation scoring across studies. Detailed postoperative complication descriptions and inclusion of patient numbers per study in Table 1 would enhance clarity. Addressing hernia defect size in both discussion and conclusion sections is crucial. Overall, despite heterogeneous data, the paper with revision can provide useful insights.

Reviewers' comments:

Reviewer's Responses to Questions

**Comments to the Author**

1. Is the manuscript technically sound, and do the data support the conclusions?

Reviewer #1: Yes

Reviewer #2: Yes

Reviewer #3: Yes

Reviewer #4: Yes

Reviewer #5: Yes

2. Has the statistical analysis been performed appropriately and rigorously? 

Reviewer #1: Yes

Reviewer #2: Yes

Reviewer #3: Yes

Reviewer #4: Yes

Reviewer #5: Yes

3. Have the authors made all data underlying the findings in their manuscript fully available?

Reviewer #1: Yes

Reviewer #2: Yes

Reviewer #3: Yes

Reviewer #4: Yes

Reviewer #5: Yes

4. Is the manuscript presented in an intelligible fashion and written in standard English?

Reviewer #1: Yes

Reviewer #2: Yes

Reviewer #3: Yes

Reviewer #4: Yes

Reviewer #5: Yes

5. Review Comments to the Author

Reviewer #1: The authors have done meta analysis of Non-fixation versus fixation of mesh in laparoscopic transabdominal preperitoneal repair of inguinal hernia

The manuscript is well written with some grammar corrections at certain places

References need to be modified as per journal criteria

figures and tables adequate

Reviewer #2: Dear Author,

The paper is lucidly written and I enjoyed reading it, except that some subheadings are starting with small letters. Also, few reference may need formatting according to the specifications. Kindly correct few formatting and grammatical mistakes. All the very best.

Thank you

Reviewer #3: Fifteen cases were converted from Laproscopy to open. How were the statistical calculations done, in view of this observation? Were the scoring for pain evaluation identical for all studies? Individual Post operative complication description in detail would be more beneficial

Reviewer #4: Thanks to the Authors for this interesting and well conducted paper.

I would like to make the following considerations.

It could be useful to add in table 1 the number of patients for each study.

The third included study, Kalidarei B et al, has patients with recurrent inguinal hernia, please could the Authors briefly specify whether after surgery with (open, TEP or TAPP) or without mesh. Moreover, were these patients excluded from recurrence rate analysis?

Although only two studies detailed the size of hernia defect, as properly reported in the discussion (line 368-370), the same clarification should be used in the conclusion when the Authors assert that non-fixation “does not increase recurrence rate”.

Reviewer #5: Your paper is interesting and well written. Unfortunately, as you highlighted, data from Literature are heterogeneous and for several aspects is difficult to draw a conclusion. Your efforts are appreciable and the results, although partial, are interesting.

6. PLOS authors have the option to publish the peer review history of their article (what does this mean?). If published, this will include your full peer review and any attached files.

Reviewer #1: **Yes: **Prof Amit Gupta

Reviewer #2: **Yes: **Dr Megha Tandon

Reviewer #3: **Yes: **Veena KL Karanth

Reviewer #4: **Yes: **Andrea Avanzolini, MD

Reviewer #5: No

---

## [Author Response · Author response to Decision Letter 0]

29 Aug 2024

Response to reviewers:

Reviewer 1：

Reviewer point #1: References need to be modified as per journal criteria.

Author response #1: Thanks for your instruction. We have checked and modified the format of reference to meet journal standards.

Reviewer 2：

Reviewer point #1: The paper is lucidly written and I enjoyed reading it, except that some subheadings are starting with small letters. Also, few reference may need formatting according to the specifications. 

Author response #1: Thank you for your comment. And we apologize for our carelessness. These errors have been modified in the revised manuscript (Line 227, Line 253, Line 260, Line 263). The references had been checked and modified.

Reviewer 3：

Reviewer point #1: Fifteen cases were converted from Laproscopy to open. How were the statistical calculations done, in view of this observation?

Author response #1: Thank you for your careful review. We checked the original study. The fifteen cases which were converted from Laproscopy to open were all included in statistical calculations for outcomes such as recurrence, seroma formation.

Reviewer point #2: Were the scoring for pain evaluation identical for all studies?

Author response #2: Thank you for your careful review. The scoring for pain evaluation was identical in five included studies. We have added the detailed description of pain evaluation in outcomes section (Line 213-216).

Reviewer point #3: Individual Post operative complication description in detail would be more beneficial.

Author response #3: Thank you for your instructive suggestion. We have added individual postoperative complication descriptions in Table 2.

Reviewer 4：

Reviewer point #1: It could be useful to add in table 1 the number of patients for each study.

Author response #1: Thank you for your instructive suggestion. We have added the patient number of each study in Table 1.

Reviewer point #2: The third included study, Kalidarei B et al, has patients with recurrent inguinal hernia, please could the Authors briefly specify whether after surgery with (open, TEP or TAPP) or without mesh. Moreover, were these patients excluded from recurrence rate analysis?

Author response #2: Thank you for your careful review. We checked the original study. We had made a mistake that the study of Kalidarei et al. [1] did not include patients of recurrent inguinal hernia. Their criteria for inclusion were “patients within the age range of 18–50 years, those with inguinal hernia, those with no history of inguinal hernial repair surgery, and those with laparotomy and strangulated or incarcerated hernia participated in the study”. We are very sorry for this careless mistake. We have made modification in the revised manuscript (Line 163-164, Table 1).

[1] Kalidarei B, Mahmoodieh M, Sharbu Z. Comparison of mesh fixation and nonfixation in laparoscopic transabdominal preperitoneal repair of inguinal hernia. Formosan Journal of Surgery 2019;52: 212-220.

Reviewer point #3: Although only two studies detailed the size of hernia defect, as properly reported in the discussion (line 368-370), the same clarification should be used in the conclusion when the Authors assert that non-fixation “does not increase recurrence rate”.

Author response #3: Thank you for your instruction. We have modified the conclusion to make it more accurate (Line 64-65, Line 379-381).

Reviewer 5:

Reviewer point #1: Your paper is interesting and well written. Unfortunately, as you highlighted, data from Literature are heterogeneous and for several aspects is difficult to draw a conclusion. Your efforts are appreciable and the results, although partial, are interesting.

Author response #1: Thank you for your comment. The heterogeneity of the data from the included studies did cause difficulties in synthetic analysis. We tried our best to make accurate conclusions, which may be interesting for surgeons in this field.

Academic Editor:

Reviewer point #1: Statistical methods need clarification regarding cases converted to open surgery, and consistency in pain evaluation scoring across studies.

Author response #1: Thank you for your careful instruction. We reviewed the study conducted by Habeeb et al. [2] carefully. They provided the causes of converting to open surgery (adhesion in the pelvis or endometriosis in the pelvis). However, they did not provide any other details of these cases, such as which study group they came from. The fifteen cases which were converted from Laproscopy to open were all included in statistical calculations for outcomes of that study. We have modified the description of pain evaluation scoring across studies in outcomes section (Line 213-216) to clarify the consistency. Four studies with available pain score data all used the same Visual Analog Scale (VAS) pain scale.

[2] Habeeb T, Mokhtar MM, Sieda B, Osman G, Ibrahim A, Metwalli AM, et al. Changing the innate consensus about mesh fixation in trans-abdominal preperitoneal laparoscopic inguinal hernioplasty in adults: Short and long term outcome. Randomized controlled clinical trial. Int J Surg 2020;83: 117-124.

Reviewer point #2: Detailed postoperative complication descriptions and inclusion of patient numbers per study in Table 1 would enhance clarity.

Author response #2: Thank you for your instructive suggestion. We have added the patient numbers for each study in Table 1. Considering the gist of each table, we have added postoperative complication descriptions in Table 2. 

Reviewer point #3: Addressing hernia defect size in both discussion and conclusion sections is crucial.

Author response #3: Thank you for your instruction. Hernia defect size is an important parameter for evaluating inguinal hernia. It is always taken into account in the treatment of hernias. Unfortunately, only two of the studies in our meta-analysis reported hernia defect size. This is indeed a limitation of our study. We have modified the conclusion to make it more accurate (Line 64-65, Line 379-381).

---

## [Decision Letter · Decision Letter 1]

11 Nov 2024

Non-fixation versus fixation of mesh in laparoscopic transabdominal preperitoneal repair of inguinal hernia: a systematic review and meta-analysis of randomized controlled trials

PONE-D-24-23572R1

Dear Dr. bai,

We’re pleased to inform you that your manuscript has been judged scientifically suitable for publication and will be formally accepted for publication once it meets all outstanding technical requirements.

Kind regards,

Antoine Naem, M.D.

Academic Editor

PLOS ONE

Additional Editor Comments (optional):

Reviewers' comments:

Reviewer's Responses to Questions

**Comments to the Author**

1. If the authors have adequately addressed your comments raised in a previous round of review and you feel that this manuscript is now acceptable for publication, you may indicate that here to bypass the “Comments to the Author” section, enter your conflict of interest statement in the “Confidential to Editor” section, and submit your "Accept" recommendation.

Reviewer #1: All comments have been addressed

Reviewer #2: All comments have been addressed

Reviewer #4: All comments have been addressed

2. Is the manuscript technically sound, and do the data support the conclusions?

Reviewer #1: Yes

Reviewer #2: Yes

Reviewer #4: Yes

3. Has the statistical analysis been performed appropriately and rigorously? 

Reviewer #1: Yes

Reviewer #2: N/A

Reviewer #4: Yes

4. Have the authors made all data underlying the findings in their manuscript fully available?

Reviewer #1: Yes

Reviewer #2: Yes

Reviewer #4: Yes

5. Is the manuscript presented in an intelligible fashion and written in standard English?

Reviewer #1: Yes

Reviewer #2: Yes

Reviewer #4: Yes

6. Review Comments to the Author

Reviewer #1: All the suggestions have been incorporated by the authors

The tables have been modified as per suggestions

References have been modified as per journal criteria

Reviewer #2: (No Response)

Reviewer #4: Thanks to the Authors for taking into consideration the comments and for the efforts in order to enhance the paper.

7. PLOS authors have the option to publish the peer review history of their article (what does this mean?). If published, this will include your full peer review and any attached files.

Reviewer #1: **Yes: **Amit Gupta

Reviewer #2: **Yes: **Dr Megha Tandon

Reviewer #4: **Yes: **Andrea Avanzolini, MD

---

## [Editor Report · Acceptance letter]

27 Nov 2024

PONE-D-24-23572R1 

PLOS ONE

Dear Dr. Bai, 

I'm pleased to inform you that your manuscript has been deemed suitable for publication in PLOS ONE. Congratulations! Your manuscript is now being handed over to our production team.

Kind regards, 

on behalf of

Dr. Antoine Naem 

Academic Editor

PLOS ONE